# Multi-Layered Gradient Boosting Decision Trees

**Ji Feng†, ‡, Yang Yu†, Zhi-Hua Zhou†**
†National Key Lab for Novel Software Technology, Nanjing University, China
†{fengj, yuy, zhouzh}@lamda.nju.edu.cn
‡ Sinovation Ventures AI Institute
‡{fengji}@chuangxin.com

## Abstract

Multi-layered distributed representation is believed to be the key ingredient of deep neural networks especially in cognitive tasks like computer vision. While non-differentiable models such as gradient boosting decision trees (GBDTs) are still the dominant methods for modeling discrete or tabular data, they are hard to incorporate with such representation learning ability. In this work, we propose the multi-layered GBDT forest (mGBDTs), with an explicit emphasis on exploring the ability to learn hierarchical distributed representations by stacking several layers of regression GBDTs as its building block. The model can be jointly trained by a variant of target propagation across layers, without the need to derive back-propagation nor differentiability. Experiments confirmed the effectiveness of the model in terms of performance and representation learning ability.

## 1 Introduction

The development of deep neural networks has achieved remarkable advancement in the field of machine learning during the past decade. By constructing a hierarchical or "deep" structure, the model is able to learn good representations from raw data in both supervised and unsupervised settings which is believed to be its key ingredient. Successful application areas include computer vision, speech recognition, natural language processing and more [Goodfellow *et al.*, 2016].

Currently, almost all the deep neural networks use back-propagation [Werbos, 1974; Rumelhart *et al.*, 1986] with stochastic gradient descent as the workhorse behind the scene for updating parameters during training. Indeed, when the model is composed of differentiable components (e.g., weighted sum with non-linear activation functions), it appears that back-prop is still currently the best choice. Some other methods such as target propagation [Bengio, 2014] has been proposed as an alternative for training, the effectiveness and popularity are however still in a premature stage. For instance, the work in Lee *et al.* [2015] proved that target propagation can be at most as good as back-prop, and in practice an additional back-propagation for fine-tuning is often needed. In other words, the good-old back-propagation is still the most effective way to train a differentiable learning system such as neural networks.

On the other hand, the need to explore the possibility to build a multi-layered or deep model using non-differentiable modules is not only of academic interest but also with important application potentials. For instance, tree-based ensembles such as Random Forest [Breiman, 2001] or gradient boosting decision trees (GBDTs) [Friedman, 2000] are still the dominant way of modeling discrete or tabular data in a variety of areas, it thus would be of great interest to obtain a hierarchical distributed representation learned by tree ensembles on such data. In such cases, there is no chance to use chain rule to propagate errors, thus back-propagation is no longer possible. This yields to two fundamental questions: First, can we construct a multi-layered model with non-differentiable components, such that the outputs in the intermediate layers are distributed representations? Second, if so, how to jointly

train such models without the help of back-propagation? The goal of this paper is to provide such an attempt.

Recently Zhou and Feng [2017; 2018] proposed the Deep Forest framework, which is the first attempt to constructing a multi-layered model using tree ensembles. Concretely, by introducing fine-grained scanning and cascading operations, the model is able to construct a multi-layered structure with adaptive model complexity and achieved competitive performance across a board range of tasks. The gcForest model proposed in [2018] utilized all strategies for diversity enhancement of ensemble learning, however, the current approach is only suitable in a supervised learning setting. Meanwhile, it is still not clear how to construct a multi-layered model by forest that *explicitly* examine its representation learning ability. Such explorations for representation learning should be made since many previous researches have suggested that, a multi-layered *distributed representations* [Hinton *et al.*, 1986] may be the key reason for the success of deep neural networks [Bengio *et al.*, 2013a].

In this work, we aim to take the best parts of both worlds: the excellent performance of tree ensembles and the expressive power of hierarchical distributed representations (which has been mainly explored in neural networks). Concretely, we propose the first multi-layered structure using gradient boosting decision trees as building blocks per layer with an explicit emphasis on its representation learning ability and the training procedure can be jointly optimized via a variant of target propagation. The model can be trained in both supervised and unsupervised settings. This is the first demonstration that we can indeed obtain *hierarchical* and *distributed* representations using trees which was commonly believed only possible for neural networks or differentiable systems in general. Theoretical justifications as well as experimental results showed the effectiveness of this approach.

The rest of the paper is organized as follows: first, some more related works are discussed; second, the proposed method with theoretical justifications are presented; finally, empirical experiments and conclusions are illustrated and discussed.

## 2 Related Works

There is still no universal theory in explaining why a deep model works better than a shallow one. Many of the current attempts [Bengio *et al.*, 2013b,c] for this question are based on the conjecture that it is the hierarchical distributed representations learned from data are the driven forces behind the effectiveness of deep models. Similar works such as [Bengio *et al.*, 2013b] conjectured that better representations can be exploited to produce faster-mixing Markov chains, therefore, a deeper model always helps. Tishby and Zaslavsky Tishby and Zaslavsky [2015] treated the hidden layers as a successive refinement of relevant information and a deeper structure helps to speed up such process exponentially. Nevertheless, it seems for a deep model to work well, it is critical to obtain a better feature re-representation from intermediate layers.

For a multi-layered deep model with differentiable components, back-propagation is still the dominant way for training. In recent years, some alternatives have been proposed. For instance, target-propagation [Bengio, 2014] and difference target propagation [Lee *et al.*, 2015] propagate the targets instead of errors via the inverse mapping. By doing so, it helps to solve the vanishing gradient problem and the authors claim it is a more biologically plausible training procedure. Similar approaches such as feedback-alignment [Lillicrap *et al.*, 2016] used asymmetric feedback connections during training and direct feedback alignment [Nøkland, 2016] showed it is possible when the feedback path is disconnected from the forward path. Currently, all these alternatives stay in the differentiable regime and their theoretical justifications depend heavily on calculating the Jacobians for the activation functions.

Ensemble learning [Zhou, 2012] is a powerful learning paradigm which often uses decision trees as its base learners. Bagging [Breiman, 1996] and boosting [Freund and Schapire, 1999] , for instance, are the driven forces of Random Forest [Breiman, 2001] and gradient boosting decision trees [Friedman, 2000], respectively. In addition, some efficient implementations for GBDTs such as XGBoost [Chen and Guestrin, 2016] and LightGBM [Ke *et al.*, 2017] has become the best choice for many industrial applications and data science projects, ranging from predicting clicks on Ads [He *et al.*, 2014], to discovering Higgs Boson [Chen and He, 2015] and numerous data science competitions in Kaggle[1]

and beyond. Some more recent works such as eForest [Feng and Zhou, 2018] showed the possibility to recover the input pattern with almost perfect reconstruction accuracy by forest. Due to the unique property of decision trees, such models are naturally suitable for modeling discrete data or data sets with mixed-types of attributes. There are some works tries to combine the routing structure of trees with neural networks [Kontschieder *et al.*, 2015; Frosst and Hinton, 2017], however, these approaches require heavily on the differential property for the system and thus are quite different with our purpose and motivation.

# 3 The Proposed Method

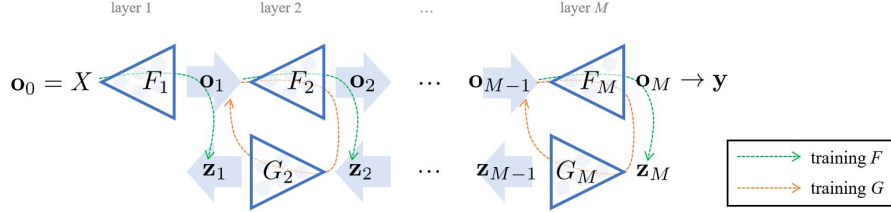

Figure 1: Illustration of training mGBDTs

Consider a multi-layered feed-forward structure with $M - 1$ intermediate layers and one final output layer. Denote $\mathbf{o}_i$ where $i \in \{0, 1, 2, \ldots, M\}$ as the output for each layer including the input layer and the output layer $\mathbf{o}_M$. For a particular input data $\mathbf{x}$, the corresponding output at each layer is in $R^{d_i}$, where $i \in \{0, 1, 2, \ldots, M\}$. The learning task is therefore to learn the mappings $F_i : R^{d_{i-1}} \to R^{d_i}$ for each layer $i > 0$, such that the final output $\mathbf{o_M}$ minimize the empirical loss $L$ on training set. Mean squared errors or cross-entropy with extra regularization terms are some common choices for the loss $L$. In an unsupervised setting, the desired output $Y$ can be the training data itself, which leads to an auto-encoder and the loss function is the reconstruction errors between the output and the original input.

When each $F_i$ is parametric and differentiable, such learning task can be achieved in an efficient way using back-propagation. The basic routine is to calculate the gradients of the loss function with respect to each parameter at each layer using the chain rule, and then perform gradient descent for parameter updates. Once the training is done, the output for the intermediate layers can be regarded as the new representation learned by the model. Such hierarchical dense representation can be interpreted as a multi-layered abstraction of the original input and is believed to be critical for the success of deep models.

However, when $F_i$ is non-differentiable or even non-parametric, back-prop is no longer applicable since calculating the derivative of loss function with respect to its parameters is impossible. The rest of this section will focus on solving this problem when $F_i$ are gradient boosting decision trees.

First, at iteration $t$, assume the $F_i^{t-1}$ obtained from the previous iteration are given, we need to obtain an "pseudo-inverse" mapping $G_i^t$ paired with each $F_i^{t-1}$ such that $G_i^t(F_i^{t-1}(\mathbf{o}_{i-1})) \approx \mathbf{o}_{i-1}$. This can be achieved by minimizing the expected value of the reconstruction loss function as: $\hat{G}_i^t = \arg\min_{G_i^t} \mathbb{E}_x[L^{inverse}(\mathbf{o}_{i-1}, G_i^t(F_i^{t-1}(\mathbf{o}_{i-1})))]$ ,where the loss $L^{inverse}$ can be the reconstruction loss. Like an autoencoder, random noise injection is often suggested, that is, instead of using a pure reconstruction error measure, it is good practice setting $L^{inverse}$ as: $L^{inverse} = \|G_i(F_i(\mathbf{o}_{i-1} + \varepsilon)) - (\mathbf{o}_{i-1} + \varepsilon)\|, \varepsilon \sim \mathcal{N}(\mathbf{0}, diag(\sigma^2))$. By doing so, the model is more robust in the sense that the inverse mapping is forced to learn how to map the neighboring training data to the right manifold. In addition, such randomness injection also helps to design a generative model by treating the inverse mapping direction as a generative path which can be considered as future works for exploration.

Second, once we updated $G_i^t$, we can use it as given and update the forward mapping for the previous layer $F_{i-1}$. The key here is to assign a pseudo-labels $\mathbf{z}_{i-1}^t$ for $F_{i-1}$ where $i \in \{2, ..M\}$, and each layer's pseudo-label is defined to be $\mathbf{z}_{i-1}^t = G_i(\mathbf{z}_i^t)$. That is, at iteration $t$, for all the intermediate layers, the pseudo-labels for each layer can be "aligned" and propagated from the output layer to the input layer. Then, once the pseudo-labels for each layer is computed, each $F_i^{t-1}$ can follow a

gradient ascent step towards the pseudo-residuals $-\frac{\partial L(F_i^{t-1}(\mathbf{o}_{i-1}), \mathbf{z}_i^t)}{\partial F_i^{t-1}(\mathbf{o}_{i-1})}$ just like a typical regression GBDT.

The only thing remains is to set the pseudo-label $\mathbf{z}_M^t$ for the final layer to make the whole structure ready for update. It turns out to be easy since at layer $M$, one can always use the real labels $y$ when defining the output layer's pseudo-label. For instance, it is natural to define the pseudo-label of the output layer as: $\mathbf{z}_M^t = \mathbf{o}_M - \alpha\frac{\partial L(\mathbf{o}_M, y)}{\partial \mathbf{o}_M}$. Then, $F_M^t$ is set to fit towards the pseudo-residuals $-\frac{\partial L(F_M^{t-1}(\mathbf{o}_{M-1}), \mathbf{z}_M^t)}{\partial F_M^{t-1}(\mathbf{o}_{M-1})}$. In other words, at iteration $t$, the output layer $F_M$ compute its pseudo-label $\mathbf{z}_M^t$ and then produce the pseudo-labels for all the other layer via the inverse functions, then each $F_i$ can thus be updated accordingly. Once all the $F_i$ get updated, the procedure can then move to the next iteration to update $G_i$. In practice, a bottom up update is suggested (update $F_i$ before $F_j$ for $i < j$) and each $F_i$ can go several rounds of additive boosting steps towards its current pseudo-label.

When training a neural network, the initialization can be achieved by assigning random Gaussian noise to each parameter, then the procedure can move on to the next stage for parameter update. For tree-structured model described here, it is not a trivial task to draw a random tree structure from the distribution of all the possible tree configurations, therefore instead of initializing the tree structure at random, we produce some Gaussian noise to be the output of intermediate layers and train some very tiny trees to obtain $F_i^0$, where index $0$ denote the tree structures obtained in this initialization stage. Then the training procedure can move on to iterative update forward mappings and inverse mappings. The whole procedure is summarized in Algorithm 1 and illustrated in Figure 1.

---

**Algorithm 1:** Training multi-layered GBDT (mGBDT) Forest

**Input:** Number of layers $M$, layer dimension $d_i$, training data $X, Y$, final loss function $L$, $\alpha, \gamma, K_1, K_2$, epoch $E$, noise injection $\sigma^2$

**Output:** A trained mGBDT

$F_{1:M}^0 \leftarrow$ Initialize(); $G_{2:M}^0 \leftarrow$ Initialize(); $\mathbf{o}_0 \leftarrow X$; $\mathbf{o}_j \leftarrow F_j^0(\mathbf{o}_{j-1})$ for $j = 1, 2, \ldots, M$

**for** $t = 1$ *to* $E$ **do**

$\quad \mathbf{z}_M^t \leftarrow \mathbf{o}_M - \alpha\frac{\partial L(\mathbf{o}_M, Y)}{\partial \mathbf{o}_M}$ // Calculate the pseudo-label for the final layer

$\quad$ **for** $j = M$ *to* $2$ **do**

$\quad\quad G_j^t \leftarrow G_j^{t-1}$

$\quad\quad \mathbf{o}_{j-1}^{noise} \leftarrow \mathbf{o}_{j-1} + \varepsilon, \varepsilon \sim \mathcal{N}(\mathbf{0}, diag(\sigma^2))$

$\quad\quad$ **for** $k = 1$ *to* $K_1$ **do**

$\quad\quad\quad L_j^{inv} \leftarrow \|G_j^t(F_j^{t-1}(\mathbf{o}_{j-1}^{noise})) - \mathbf{o}_{j-1}^{noise}\|$

$\quad\quad\quad \mathbf{r}_k \leftarrow -[\frac{\partial L_j^{inv}}{\partial G_j^t(F_j^{t-1}(\mathbf{o}_{j-1}^{noise}))}]$

$\quad\quad\quad$ Fit regression tree $h_k$ to $\mathbf{r}_k$, i.e. using the training set $(F_j^{t-1}(\mathbf{o}_{j-1}^{noise}), \mathbf{r}_k)$

$\quad\quad\quad G_j^t \leftarrow G_j^t + \gamma h_k$

$\quad\quad$ **end**

$\quad\quad \mathbf{z}_{j-1}^t \leftarrow G_j^t(\mathbf{z}_j^t)$ // Calculate the pseudo-label for layer $j - 1$

$\quad$ **end**

$\quad$ **for** $j = 1$ *to* $M$ **do**

$\quad\quad F_j^t \leftarrow F_j^{t-1}$

$\quad\quad$ // Update $F_j^t$ using pseudo-label $\mathbf{z}_j^t$ for $K_2$ rounds

$\quad\quad$ **for** $k = 1$ *to* $K_2$ **do**

$\quad\quad\quad L_j \leftarrow \|F_j^t(\mathbf{o}_{j-1}) - \mathbf{z}_j^t\|$

$\quad\quad\quad \mathbf{r}_k \leftarrow -[\frac{\partial L_j}{\partial F_j^t(\mathbf{o}_{j-1})}]$

$\quad\quad\quad$ Fit regression tree $h_k$ to $\mathbf{r}_k$, i.e. using the training set $(\mathbf{o}_{j-1}, \mathbf{r}_k)$

$\quad\quad\quad F_j^t \leftarrow F_j^t + \gamma h_k$

$\quad\quad$ **end**

$\quad\quad \mathbf{o}_j \leftarrow F_j^t(\mathbf{o}_{j-1})$

$\quad$ **end**

**end**

return $F_{1:M}^T, G_{2:M}^T$

---

It is worth noticing that the work in Rory and Eibe [2017] utilized GPUs to speed up the time required to train a GBDT and Korlakai and Ran [2015] showed an efficient way of conducting drop-out techniques for GBDTs which will give a performance boost further. For a multi-dimensional output problem, the naïve approaches using GBDTs would be memory inefficient and Si *et al.* [2017] proposed an efficient way of solving such problem which can reduce the memory by an order of magnitude in practice.

In classification tasks, one could set the forward mapping in the output layer as a linear classifier. There are two main reasons of doing this: First, by doing so, the lower layers will be forced to learn a feature re-representation that is as linear separable as possible which is a useful property to have. Second, often the difference of the dimensionality between the output layer and the layer below is big, as a result, an accurate inverse mapping may be hard to learn. When using a linear classifier as the forward mapping at the output layer, there is no need to calculate that particular corresponding inverse mapping since the pseudo-label for the layer below can be calculated by taking the gradient of the global loss with respect to the output of the last hidden layer.

A similar procedure such as target propagation [Bengio, 2014] has been proposed to use the inter-layer feedback mappings to train a neural network. They proved that under certain conditions, the angle between the update directions for the parameters of the forward mappings and the update directions when trained with back-propagation is less than 90 degree. However, the proof relies heavily on the computing the Jacobians of $F_i$ and $G_i$, therefore, their results are only suitable for neural networks.

The following theorem proves that, under certain conditions, an update in the intermediate layer towards its pseudo-label helps to reduce the loss of the layer above, and thus helps to reduce the global loss. The proof here does not rely on the differentiable property of $F_i$ and $G_i$.

**Theorem 1.** *Denote an update of $f_{i-1}^{old}$ to $f_{i-1}^{new}$ moves its output from $h_i$ to $h_i'$, where $h_i$ and $h_i'$ are in $R^{d_i}$ and denote the input for $f_{i-1}$ as $h_{i-1}$ which is in $R^{d_{i-1}}$. Assume each $f_i$ is t-Lipchize continuous on $R^{di}$ and $g_i = f_i^{-1}$ is 1/t-Lipchize continuous.[2]. Now suppose such update for $f_{i-1}$ reduced its local loss, that is, $\|f_{i-1}^{new}(h_{i-1}) - Target_{i-1}\| \le \|f_{i-1}^{old}(h_{i-1}) - Target_{i-1}\|$, then it helps to reduce the loss for the layer above, that is, the following holds:*

$$\|f_i(h_i') - Target_i\| \le \|f_i(h_i) - Target_i\| \tag{1}$$

*Proof.* By assumption, it is easy to show that $\|f_i^{-1}f_i(x) - f_i^{-1}f_i(y)\| \le \|f_i(x) - f_i(y)\|/t$ and $\|g_i^{-1}g_i(x) - g_i^{-1}g_i(y)\| \le t\|g_i(x) - g_i(y)\|$. Then we have the following:

$$
\begin{aligned}
\|f_i(h_i') - Target_i\| &\le t\|g_i(f_i(h_i')) - g_i(Target_i)\| \\
&= t\|h_i' - Target_{i-1}\| \\
&= t\|f_{i-1}^{new}(h_{i-1}) - Target_{i-1}\| \\
&\le t\|f_{i-1}^{old}(h_{i-1}) - Target_{i-1}\| \\
&\le t\|f_i(f_{i-1}^{old}(h_{i-1})) - f_i(Target_{i-1})\|/t \\
&= \|f_i(f_{i-1}^{old}(h_{i-1})) - f_i(Target_{i-1})\| \\
&= \|f_i(h_i) - Target_i\|
\end{aligned}
$$

$\square$

To conclude this section, here we discuss several reasons for the need to explore non-differential components in designing multi-layered models. Firstly, current adversarial attacks [Nguyen *et al.*, 2015; Huang *et al.*, 2017] are all based on calculating the derivative of the final loss with respect to the input. That is, *regardless* of the training procedure, one can always attack the system as long as chain rule is applicable. Non-differentiable modules such as trees can naturally block such calculation, therefore, it would more difficult to perform malicious attacks. Secondly, there are still numerous data sets of interest that are best suitable to be modeled by trees. It would be of great interests and

potentials to come up with algorithms that can blend the performance of tree ensembles with the benefit of having a multi-layered representation.

## 4 Experiments

The experiments for this section is mainly designed to empirically examine if it is feasible to jointly train the multi-layered structure proposed by this work. That is, we make no claims that the current structure can outperform CNNs in computer vision tasks. More specifically, we aim to examine the following questions: **(Q1)** Does the training procedure empirically converge? **(Q2)** What does the learned features look like? **(Q3)** Does depth help to learn a better representation? **(Q4)** Given the same structure, what is the performance compared with neural networks trained by either back-propagation or target-propagation? With the above questions in mind, we conducted 3 sets of experiments with both synthetic data and real-world applications which results are presented below.

### 4.1 Synthetic Data

As a sanity check, here we train two small multi-layered GBDTs on synthetic datasets.

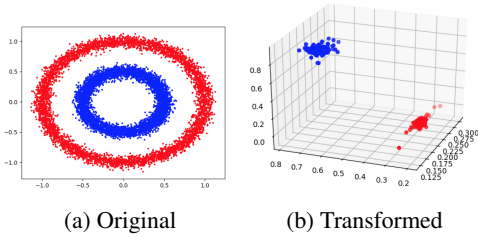

(a) Original　　　(b) Transformed

Figure 2: Supervised classification

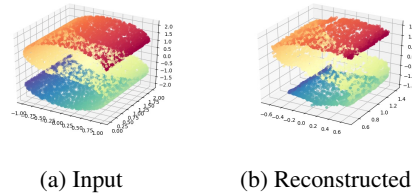

(a) Input　　　(b) Reconstructed

Figure 3: Unsupervised mGBDT autoencoder

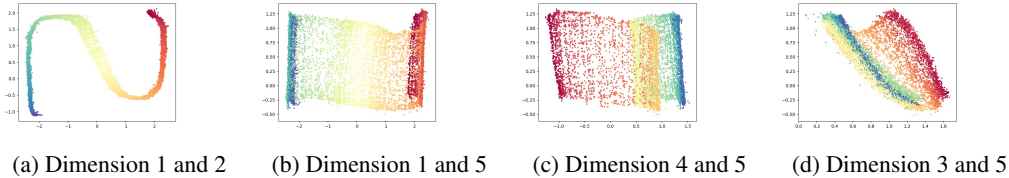

(a) Dimension 1 and 2　　(b) Dimension 1 and 5　　(c) Dimension 4 and 5　　(d) Dimension 3 and 5

Figure 4: Visualizations in the $5D$ encoding space of unsupervised mGBDT autoencoder

We generated $15,000$ points with 2 classes (70 % for training and 30 % for testing) on $R^2$ as illustrated in Figure 2a. The structure we used for training is $(input - 5 - 3 - output)$ where the input points are in $R^2$ and the output is a 0/1 classification prediction. The mGBDT used in both forward and inverse mappings have a maximum depth of 5 per tree with learning rate of 0.1. The output of the last hidden layer (which is in $R^3$) is visualized in Figure 2b. Clearly, the model is able to transform the data points that is easier to separate.

We also conducted an unsupervised learning task for autoencoding. $10,000$ points in $R^3$ with shape $S$ were generated, as shown in Figure 3a. Then we built an autoencoder using mGBDTs with structure $(input - 5 - output)$ with MSE as its reconstruction loss. The hyper-parameters for tree configurations are the same as the 2-class classification task. In other words, the model is forced to learn a mapping from $R^3$ to $R^5$, then maps it back to the original space with low reconstruction error as objective. The reconstructed output is presented in Figure 3b. The $5D$ encodings for the input $3D$ points are impossible to visualize directly, here we use a common strategy to visualize some *pairs* of dimensions for the $5D$ encodings in $2D$ as illustrated in Figure 4. The $5D$ representation for the $3D$ points is indeed a distributed representation [Hinton *et al.*, 1986] as some of the dimension captures the curvature whereas others preserve the relative distance among points.

### 4.2 Income Prediction

The income prediction dataset [Lichman, 2013] consists of $48,842$ samples ($32,561$ for training and $16,281$ for testing) of tabular data with both categorical and continuous attributes. Each sample

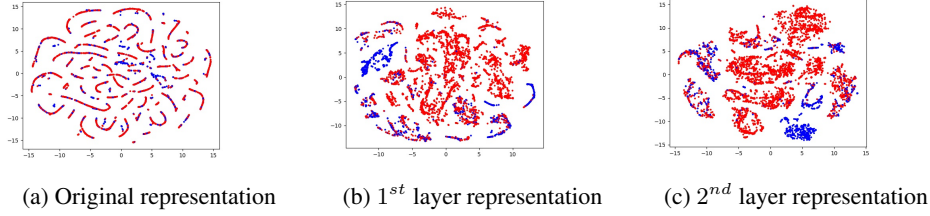

(a) Original representation      (b) $1^{st}$ layer representation      (c) $2^{nd}$ layer representation

Figure 5: Feature visualization for income dataset

consists of a person's social background such as race, sex, work-class, etc. The task here is to predict whether this person makes over 50K a year. One-hot encoding for the categorical attributes make each training data in $R^{113}$. The multi-layered GBDT structure we used is $(input - 128 - 128 - output)$. Gaussian noise with zero mean and standard deviation of 0.3 is injected in $L^{inverse}$. To avoid training the inverse mapping on the output layer, we set the final output layer to be a linear with cross-entropy loss, other layers all use GBDTs for for forward/inverse mappings with the same hyper-parameters in section 4.1. The learning rate $\alpha$ at output layer is determined by cross-validation. The output for each intermediate layers are visualized via T-SNE [van der Maaten and Hinton, 2008] in Figure 5. We wish to highlight that all the mGBDTs used exactly the same hyper-parameters across all the experiments: 5 additive trees per epoch ($K_1 = K_2 = 5$), the maximum depth is fixed to be 5. Such rule-of-thumb setting is purposely made in order to avoid a fine-tuned performance report.

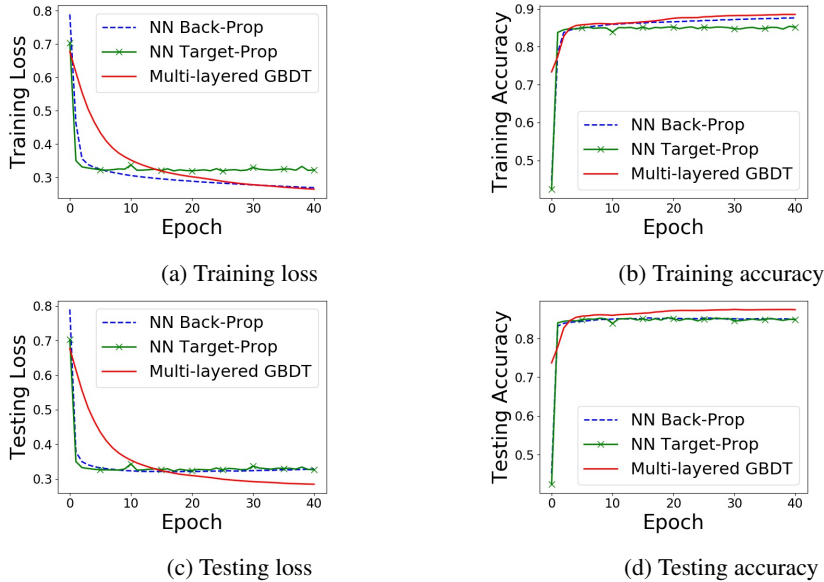

(a) Training loss            (b) Training accuracy

(c) Testing loss            (d) Testing accuracy

Figure 6: Learning curves of income dataset

For a comparison, we also trained the exact same structure $(input - 128 - 128 - output)$ on neural networks using the target propagation $NN^{TargetProp}$ and standard back-propagation $NN^{BackProp}$, respectively. Since the goal here is to compare the predictive accuracy given the same representational dimensions therefore other NN architectures are not reported in details. (Actually smaller NNs won't help, for instance, $(input - 32 - 32 - output)$ achieved 85.20% and $(input - 16 - 16 - output)$ achieved 84.67%.) Adam [Kingma and Ba, 2014] with a learning rate of 0.001 and ReLU activation are used for both cases. Dropout rate of 0.25 is used for back-prop. A vanilla XGBoost via cross-validation search for hyper-parameters with 100 additive trees with a maximum depth of 7 per tree is also trained for comparison, the optimal learning rate found for XGBoost is 0.3. Finally, we stacked 3 XGBoost of the same configurations as the vanilla XGBoost and used one additional XGBoost as the second stage of stacking via 3-fold validation. More stacking levels will produce severe over-fitting results and are not included here.

Experimental results are summarized in Figure 6 and Table 1. First, multi-layered GBDT forest (mGBDT) achieved the highest accuracy compared to DNN approaches trained by either back-prop or target-prop, given the same model structure. It also performs better than single GBDTs or stacking multiple ones in terms of accuracy. Second, $NN^{TargetProp}$ converges not as good as $NN^{BackProp}$ as expected (a consistent result with Lee *et al.* [2015]), whereas the same structure using GBDT layers can achieve a lower training loss without over-fitting.

Table 1: Classification accuracy comparison. For protein dataset, accuracy measured by 10-fold cross-validation shown in mean $\pm$ std.

|  | Income Dataset | Protein Dataset |
|---|---|---|
| XGBoost | .8719 | .5937 $\pm$ .0324 |
| XGBoost Stacking | .8697 | .5592 $\pm$ .0400 |
| $NN^{TargetProp}$ | .8491 | .5756 $\pm$ .0465 |
| $NN^{BackProp}$ | .8534 | .5907 $\pm$ .0268 |
| Multi-layered GBDT | **.8742** | **.5948 $\pm$ .0268** |

## 4.3 Protein Localization

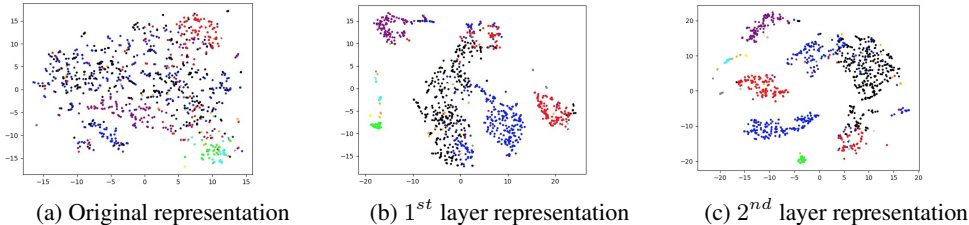

(a) Original representation     (b) $1^{st}$ layer representation     (c) $2^{nd}$ layer representation

Figure 7: Feature visualization for protein dataset

The protein dataset [Lichman, 2013] is a 10 class classification task consists of only 1484 training data where each of the 8 input attributes is one measurement of the protein sequence, the goal is to predict protein localization sites with 10 possible choices. 10-fold cross-validation is used for model evaluation since there is no test set provided. We trained a multi-layered GBDT using structure $(input - 16 - 16 - output)$. Due to the robustness of tree ensembles, all the training hyper-parameters are the same as we used in the previous section. Likewise, we trained two neural networks $NN^{TargetProp}$ and $NN^{BackProp}$ with the same structure, and the training parameters were determined by cross-validation for a fair comparison. Experimental results are presented in Table 1. Again mGBDT achieved best performance among all. XGBoost Stacking had a worse accuracy than using a single XGBoost, this is mainly because over-fitting has occurred. We also visualized the output for each mGBDT layer using T-SNE in Figure 7. It can be shown that the quality of the representation does get improved with model depth.

The training and testing curves for 10-fold cross-validation are plotted with mean value in Figure 8. The multi-layered GBDT (mGBDT) approach converges much faster than NN approaches in terms of number of epochs, as illustrated in Figure 8a. Only 50 epoch is needed for mGBDT whereas NNs require 200 epochs for both back-prop and target-prop scenarios. When measured by the wall-clock time, mGBDT runs close to NN (only slower by a factor of 1.2) with backprops in our experiments and mGBDT has a training speed very close to NN with target-prop. Nevertheless, comparing wall-clock time is less meaningful since mGBDT and NNs use different devices (CPU v.s. GPU) and different implementation optimizations. In addition, $NN^{TargetProp}$ is still sub-optimal than $NN^{BackProp}$ and mGBDT achieved highest accuracy among all. We also examined the effect when we vary the number of intermediate layers on protein datasets. To make the experiments manageable, the dimension for each intermediate layer is fixed to be 16. The results are summarized in Table 2. It can be shown that mGBDT is more robust compared with $NN^{TargetProp}$ as we increase the intermediate layers. Indeed, the performance dropped from .5964 to .3654 when using target-prop for neural networks whereas mGBDT can still perform well when adding extra layers.

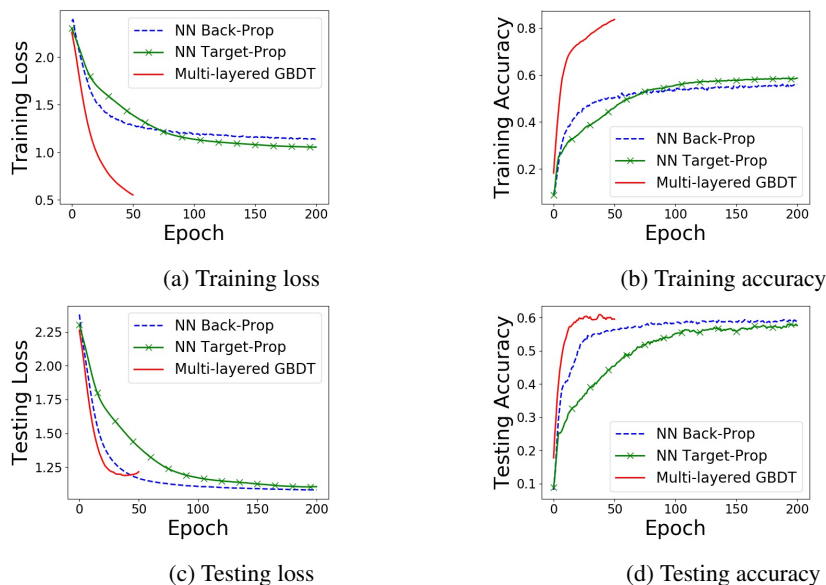

Figure 8: Learning curves of protein dataset

Table 2: Test accuracy with different model structure. Accuracy measured by 10-fold cross-validation shown in mean $\pm$ std. N/A stands for not applicable.

| Model Structure | $NN^{BackProp}$ | $NN^{TargetProp}$ | $mGBDT$ |
|---|---|---|---|
| 8->10 | $.5873 \pm .0396$ | N/A | $.5937 \pm .0324$ |
| 8->16->10 | $.5803 \pm .0316$ | $.5964 \pm .0343$ | $.6160 \pm .0323$ |
| 8->16->16->10 | $.5907 \pm .0268$ | $.5756 \pm .0465$ | $.5948 \pm .0268$ |
| 8->16->16->16->10 | $.5901 \pm .0270$ | $.4759 \pm .0429$ | $.5897 \pm .0312$ |
| 8->16->16->16->16->10 | $.5768 \pm .0286$ | $.3654 \pm .0452$ | $.5782 \pm .0229$ |

## 5 Conclusion and Future Explorations

In this paper, we present a novel multi-layered GBDT forest (mGBDT) with explicit representation learning ability that can be jointly trained with a variant of target propagation. Due to the excellent performance of tree ensembles, this approach is of great potentials in many application areas where neural networks are not the best fit. The work also showed that, to obtain a multi-layered distributed representations is not tired to differentiable systems. Theoretical justifications, as well as experimental results confirmed the effectiveness of this approach. Here we list some aspects for future explorations.

**Deep Forest Integration.** One important feature of the deep forest model proposed in [Zhou and Feng, 2018] is that the model complexity can be adaptively determined according to the input data. Therefore, it is interesting to integrating several mGBDT layers as feature extractor into the deep forest structure to make the system not only capable of learning representations but also can automatically determine its model complexity.

**Structural Variants and Hybird DNN.** A recurrent or even convolutional structure using mGBDT layers as building blocks are now possible since the training method does not making restrictions on such structural priors. Some more radical design is possible. For instance, one can embed the mGBDT forest as one or several layers into any complex differentiable system and use mGBDT layers to handle tasks that are best suitable for trees. The whole system can be jointly trained with a mixture of different training methods across different layers. Nevertheless, there are plenty of room for future explorations.

**Acknowledgments** This research was supported by NSFC (61751306), National Key R&D Program of China (2018YFB1004300) and Collaborative Innovation Center of Novel Software Technology and Industrialization.

## Footnotes

[1]www.kaggle.com

[2]Clarke inverse function theorem [Clarke, 1976] proved the existence of such $g_i$ under mild conditions on generalized derivatives of $f_i$ without loss of generality.

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
