[Reviews · NeurIPS 2018]

Reviewer 1



Short overview: Authors propose to build a neural network using gradient boosted trees as components in the layers. To train such a structure, since the gbdts are not able to propagate the gradient, they propose to use a method inspired by the target propagation: each gradient boosted trees is built to approximate a gradient of loss of prediction function and a pseudo target, with respect to the prediction function. Pseudo targets are updated at each iteration using the reverse mapping of the built tree representation and the pseudo label of the next layer. The reverse mapping can be found using the reconstruction loss. At each iteration, each layer's ensemble grows by one boosting tree Authors hint at potential applications of blocking adversarial attacks, that rely on estimating the gradients of the final loss with respect to input, which would not work for layers that can't propagate the gradients, however this direction is not explored in this paper Detailed comments: Overall, an interesting idea of co-training gbdts with nns. The idea is not new, but discarding backprop and using target propagation, that allows gbdts to do what they are best for - e.g. build trees to simulate gradients - is neat and witty. Overall, well written and well structured, but I do have some points of confusion after reading it: a) Pseudo labels are all computed from top to bottom - i am not sure where you think is the top and where the bottom is. is it to say that the first pseudo label is from the last (output layer) and then you go backward to the input layer. If yes, this top to bottom needs to be rephrased. Some people draw neural nets horizontally actually... (you included, figure 1!), but even if you draw vertically, the output is the bottom, and you go from the bottom to the top b) What is the structure of your pseudo-inverse function G_i^t? Is it essentially a matrix of dimension R^d_i x R^d_(i-1)? In your algorithm, I am not sure why you fit the regression tree and update the inverse mapping interchangeably for a number of iterations? Why not to first minimize the loss (using the tree from the previous iterations) to obtain the inverse mapping (so for a number of iterations, do a gradient descent update using the reconstruction loss) and then when it is settled, build a tree using the inverse loss and found inverse mapping, which can be used to get pseudo label at this point. (i am referring to the for loop k=1..K). Instead to update the G you make a pseudo gradient descent step using the built tree, which does not seem very intuitive to me. Plus it also seems that you discard all the trees built at this step (you use them just to update G) c )As far as i remember, target propagation works slower than backprop, I am not sure if you observed the same My main concern is the experiments and their reproducibility: the way they are described, it is not clear what the actual setup was used for their method, and if the comparison is fair (were all the competing models tuned?). I am having hard time believing that well tuned 2-hidden layer nn would underperform so badly. What do you mean for a setup of 5-3 for instance? AT first i thought those were the number of neurons in nn, but on the second thought, i think you mean 3 is the dimension of the first tree layer, 5 of the second. But how many trees are in each layer? This really needs to be clarified. Why accuracy and reconstruction loss of is not reported for the synthetic datasets. As well as metrics of at least one competing baselines, like just a nn? How are those 128-128 setups are chosen for income prediction? i think the fact that nns are trained with exactly the same structure (128-128 for a hidden layer) means you have a huge overparameterization, that's why they don't do well. The structure of the net should be just tuned to provide a fair point of comparison. The same as hyperparameters for xgboost (eg learning rate of 0.3 is default and is large, income is a simple dataset, and i expect that you just overfit at this point) Additionally, I not sure what the exact target application is? The adversarial learning is interesting, but it was not explored, and apart form this, i am struggling to imagine where an ensemble per layer should be more efficient than a tuned hidden layer. Suggestions: to facilitate the reading, it would be nice to add a short section on target propagation, since it is a backbone of your method, to facilitate the reading. You seem to have enough space for that Nits: line 185: the experiments ... is->are line 187: where did CNNs came from? i think even ur synthetic experiments dont use the convolutional layer (but i am not sure). Income prediction surely does not Table 1: are the diffs significant on 16k test dataset for income? line 270: interesting to integrate line 274: does not make I read authors rebuttal and I they addressed the majority of questions I had

Reviewer 2



Summary This paper proposes to train multiple layers of gradient boosting decision trees. The training algorithm is based on target propagation. The idea is to backpropagate errors using functions trained to inverse layers. The potential of the approach for feature extraction is illustrated on a couple of artificial datasets and its predictive performance is compared against (single layer) boosting and neural networks on 3 datasets. Evaluation The idea of building multiple layers of gradient boosting models is interesting from an intellectual point of view. From a more practical point of view however, it's not so clear whether this is a good idea. Gradient boosting models are already very strong learners that obtain very good results in many applications. It is not clear that using multiple layers of this model will bring improvement in terms of expressive power. Overall, I think the approach lacks a clear motivation, beyond the rather artificial arguments given in the paper. The training algorithm derived from target propagation makes sense, although I'm missing some discussion about the possibility to inverse gradient boosting models with another gradient boosting model. Is it feasible at all? Theorem 1 is potentially interesting but the requirement of isometry of the inverse function G seems very strong and needs to be discussed further. The illustrations on synthetic data in Section 4.1 are interesting. Experiments in Sections 4.2 and 4.3 have several limitations however. First, except for the number of layers, there is no analysis of the impact of the different method parameters. Two parameters that I think needs to be studied in depth are the number of boosting iterations (i.e., K1 and K2, whose values are not given in the paper) and the depth of these trees. Their influence should also be studied as a function of the number of layers. All experiments seem to have been performed with boosting models of depth 5. It's maybe already too high to see a benefit when stacking multiple layers of these models. Second, the comparison with XGBoost and NN is not very informative. Hyper-parameters of XGBoost are fixed and chosen arbitraly. It's not clear that it can not be improved by tuning these hyperparameters. This is important if the authors wants to show the benefit of having multiple layers of GBDT. Keeping the same structure for both NN and mGBDT is unfair as NN use weaker (linear) models in their layers. Again, it's not clear that NN can not be improved easily by increasing the number of layers. Comparing also convergence of mGBDT and NN in terms of number of epochs does not seem appropriate. My guess is that one mGBDT epoch is much slower than one epoch of NN with backprop. So, I don't see the point of concluding from the experiments that mGBDT converges much faster than NN in terms of epoch. Computing times should be used instead on the x-axis of the plots in Figure 8. Third, results in terms of predictive performance are actually disappointing. On the Income and Protein datasets, the accuracy of multi-layered GBDT is not better than the accuracy of XGBoost, suggesting that having more than 1 layer does not bring much. On the protein data, even with NN, having more than one layer does not really improve. This suggests that the chosen datasets are actually too simple to highlight the potential benefit of having multiple layers. Why not use large computer vision dataset? I understand that the proposed model can not compete with convolutional neural network but still, on such complex problems, showing the benefit of having multiple layers could be easier. The paper is not very well written. There remain many typos, omitted words, ungrammatical sentences, or unsupported statements. The writing should be improved. Minor comments: - The paper does not give clear credit to target propagation for the training algorithm. How is it similar/dissimilar to target propagation? - Page 5: "there are still numerous datasets of interests that are best suitable to be modeled by trees". What are these datasets? Please give references. - Theorem 1: "preserve the isometry of its neighbors". I don't understand this expression. What is "the isometry of its neighbors"? Update after rebuttal: Thank you for your answer. I still believe that the effect of the parameters should have been studied more thoroughly. The argument that a default setting is used "to avoid a fine-tuned performance report" and "to show the robustness for mGBDTs" is not valid in my opinion. Despite what you claim, mGBDT is also not significantly better than XGBoost on the Income and Protein datasets. 0.8719 vs 0.8742 and 0.5937 vs 0.5948 (in this latter case with a standard deviation > 0.25) is not what I call an improvement. At the present stage, the experiments in the paper do not really show the practical interest of having multiple layers of GBDT.

Reviewer 3



Summary: This paper presents a novel algorithm to learn multi-layered representations by stacking gradient boosted trees. Unlike the layer-wise learning of standard stacking(stacked generalization), the proposed algorithm is capable of jointly optimizing all layers. The main ideas is to consider vector-output gradient boosted regression trees as many-to-many mappings between layers, and also learning each layer's mapping (and reverse mapping to recover the layer's input from the output) via optimization like 'target propagation'. Like the cases of learning denoising autoencoders, each layer is learned to recover the input from the output with the corresponding inverse mapping, and this joint optimization of all layers itself is done in a manner similar to the gradient boosting (approximating the negative gradient by regression trees). This enables us to apply not only supervised learning but also unsupervised learning, and actually experiment sections include various interesting examples of representation learning. Strengths: - Deep learning by neural nets is dominating for multi-layer representation learning, but this paper demonstrates that such multi-layer representation learning is also possible by gradient boosted trees, which would be a quick and accurate everyday tool of practitioners. Weakness: - This paper's approach proposes multi-layer representation learning via gradient boosted trees, and on the top of that, linear regression/softmax regression are placed for supervised learning. But, we can do the opposite representation learning via neural nets, and on the top of that, decision trees can be used (Deep Neural Decision Forests by Kontschieder et al, ICCV 2015). Comment: - Basically I liked overall ideas. It can demonstrate the deep learning is not the only option for multi-layer representation learning. It's also nice that the input-output recovery like autoencoder would be also in a gradient descent manner similar to gradient boosting itself. Seemingly, it is simpler than the stacked trees such as Deep Forests (Zhou and Feng, IJCAI 2017). - Something is wrong in the descriptions of Algorithm 1. If we initialize G_{2:M}^0 <- null, then the first run of "G_j^t <- G_j^{t-1}" for the j=M to 2 loop would become G_M^1 <- G_M^0 = null, and thus the computation of L_j^inv (as well as the residuals r_k) can be problematic. - This is just a comment, and does not need to reflect the paper this time, but there would be several interesting points to be investigated in future. First, the presented multi-layer representation is probably way weaker than the ones possible with the current neural networks. The presented representation corresponds to the fully connected (FC) architectures of neural networks, and more flexible architectures such as CNN and RNN would not be directly possible. Given this point, it would be unclear whether we should use trees for multi-layered representation learning. At least, we could use boosted model trees having linear regression at leaves, for example. As mentioned in the weakness section, 'differentiable forests' such as Deep Neural Decision Forests was already proposed, and we can use this type of approaches with the modern neural nets for 'representation learning / feature learning' parts. What situation fits what choice would be one interesting open questions. Comments after author response: Thank you for the response. It'll be nice to have the revisions on the notations for initialization, and some discussion or mentions about the different approach to integrate tree-based and multi-layer representation learning.